# The Relationship between Hospital Volume and In-Hospital Mortality of Severely Injured Patients in Dutch Level-1 Trauma Centers

**DOI:** 10.3390/jcm10081700

**Published:** 2021-04-15

**Authors:** Charlie A. Sewalt, Esmee Venema, Erik van Zwet, Jan C. van Ditshuizen, Stephanie C. E. Schuit, Suzanne Polinder, Hester F. Lingsma, Dennis den Hartog

**Affiliations:** 1Department of Public Health, Erasmus MC Medical Center, 3015 GD Rotterdam, The Netherlands; e.venema@erasmusmc.nl (E.V.); s.polinder@erasmusmc.nl (S.P.); h.lingsma@erasmusmc.nl (H.F.L.); 2Department of Neurology, Erasmus MC Medical Center, 3015 GD Rotterdam, The Netherlands; 3Department of Biostatistics, Leiden University Medical Center, 2333 ZA Leiden, The Netherlands; e.w.van_zwet@lumc.nl; 4Trauma Center South West Netherlands, Erasmus MC Medical Center, 3015 GD Rotterdam, The Netherlands; j.vanditshuizen@erasmusmc.nl (J.C.v.D.); d.denhartog@erasmusmc.nl (D.d.H.); 5Trauma Research Unit, Department of Surgery, Erasmus MC Medical Center, 3015 GD Rotterdam, The Netherlands; 6Department of Internal Medicine, Erasmus MC Medical Center, 3015 GD Rotterdam, The Netherlands; s.schuit@erasmusmc.nl

**Keywords:** volume-outcome relationship, major trauma, centralization

## Abstract

Centralization of trauma centers leads to a higher hospital volume of severely injured patients (Injury Severity Score (ISS) > 15), but the effect of volume on outcome remains unclear. The aim of this study was to determine the association between hospital volume of severely injured patients and in-hospital mortality in Dutch Level-1 trauma centers. A retrospective observational cohort study was performed using the Dutch trauma registry. All severely injured adults (ISS > 15) admitted to a Level-1 trauma center between 2015 and 2018 were included. The effect of hospital volume on in-hospital mortality was analyzed with random effects logistic regression models with a random intercept for Level-1 trauma center, adjusted for important demographic and injury characteristics. A total of 11,917 severely injured patients from 13 Dutch Level-1 trauma centers was included in this study. Hospital volume varied from 120 to 410 severely injured patients per year. Observed mortality rates varied between 12% and 24% per center. After case-mix correction, no statistically significant differences between low- and high-volume centers were demonstrated (adjusted odds ratio 0.97 per 50 extra patients per year, 95% Confidence Interval 0.90–1.04, *p* = 0.44). The variation in hospital volume of the included Level-1 trauma centers was not associated with the outcome of severely injured patients. Our results suggest that well-organized trauma centers with a similar organization of care could potentially achieve comparable outcomes.

## 1. Introduction

Injury is the major cause of death in adults younger than 45 years of age [1]. The implementation of trauma systems and dedicated level I trauma centers in the United States and the Netherlands reduced mortality of severely injured patients, usually defined as patients with an Injury Severity Score (ISS) above 15, and improved functional outcome at discharge [2,3].

Implicit to the centralization of trauma care is the idea that increased volumes of severely injured patients lead to more experienced health care providers, which could result in improved patient outcomes. A recently published systematic review showed that higher hospital volume is associated with lower mortality in severely injured patients [4]. However, hospitals treating severely injured patients do not only differ in hospital volume. Other factors, such as variation in case mix, organization of care, facilities and geographic location could cause between-center differences. In 1999, an inclusive trauma system was implemented in the Netherlands, which divided hospitals between Level-1, Level-2 and Level-3 trauma centers [5]. Dutch Level-1 trauma centers are equipped to treat the most severely injured patients 24 h, 7 d per week, including neurotrauma patients, while Level-2 trauma centers are developed for patients with multiple and less severe injuries, and Level-3 trauma centers are developed to treat patients with isolated injuries. Thirteen Level-1 trauma centers cover the country with 17 million inhabitants. The number of severely injured patients treated by these trauma centers differs from approximately 100 to 450 per year [6]. 

In the prehospital phase, Dutch emergency service providers try to discriminate between patients with severe injury and without severe injury. Each trauma region consists of at least one Level-1 trauma center and several Level-2/3 trauma centers. Severely injured patients, patients with a predicted ISS above 15, are transported to Level-1 trauma centers directly [7]. The triage protocol used is based on the Field Triage Decision Schema established by the American College of Surgeons Committee on Trauma [7]. The Dutch Helicopter Emergency Services (HEMS) are mostly activated for patients with severe neurotrauma or cardiac arrest. 

Since some Level-1 trauma centers in other countries, like the United States, receive higher numbers of severely injured patients [8], questions about the introduction of hospital volume cut-offs were raised in order to improve patient outcomes.

Therefore, the aim of this study was to determine whether there is an association between hospital volume of severely injured patients and in-hospital mortality in Dutch Level-1 trauma centers.

## 2. Materials and Methods

### 2.1. Study Population

A retrospective observational cohort study was performed using the Dutch Trauma Registry (LTR).

The LTR registers data from all Dutch trauma patients presenting at the Emergency Department (ED) within 48 h after injury, and who are directly admitted to the hospital, referred to another hospital or died at the ED [9]. Patients that died on arrival are not included in the LTR. LTR is a well-structured nationwide trauma registry that has broad inclusion criteria. This enables studies on the burden of injury, and the quality and efficiency of the trauma care system.

For this analysis, all severely injured adult patients (ISS > 15 and age > 18 years) admitted to a Level-1 trauma center between 2015 and 2018 were included. The AIS codebook 2005 update 08 was used to calculate ISS scores [6,10]. Incoming transfers from foreign hospitals were excluded from the analysis but did count as case load. The primary outcome was in-hospital mortality.

### 2.2. Statistical Analysis

First, hospital volume was calculated for each Level-1 trauma center as the number of severely injured patients admitted to a Level-1 trauma center per three years (2015–2017). To assess the volume-outcome relationship, observed mortality rates were plotted against hospital volume for all Level-1 trauma centers. To describe patient characteristics, hospital volume was divided in low- versus high-volume centers with a cut-off of 180 severely injured patients per year. The value of 180 patients per year was chosen based on the median number of patients per year. For the analysis part, missing data were imputed using multiple imputation, assuming missingness at random. All variables, except for the outcome variable, in-hospital mortality, which had no missing values, were imputed. Continuous variables were standardized using Z-score standardization before imputation. There were five imputed datasets created which were pooled using Rubin’s Rules (Table A1 and Figure A1). A complete case analysis was done to validate our results (Appendix B). 

Subsequently, multivariable random effects logistic regression models were used to analyze the effect of hospital volume on outcome (in-hospital mortality). Hospital volume was tested for nonlinearity using splines and a likelihood ratio test. Both the unadjusted and adjusted models contained hospital volume and a random intercept for centers to adjust for between-center differences. The adjusted models were based on clinically relevant parameters and included age, sex, ISS, systolic blood pressure at ED, respiratory rate at ED, GCS at ED, prehospital intubation, ASA, penetrating injury and Abbreviated Injury Score (AIS) for head injury [11]. Age was modelled with a spline function, and an interaction term was added for the relationship between the effect of age and the effect of sex [12]. Sensitivity analyses were performed in subgroups of patients with ISS > 24, AIS Head < 4 and AIS Head > 3. All random effects logistic regression models were evaluated for multicollinearity by assessing the correlation between the included variables. A cut-off value of −0.8 or 0.8 was used as measure of severe multicollinearity. 

Statistical analyses were performed in R statistical software 3.5.1 (R Foundation for Statistical Computation, Vienna, Austria). Random effect models were fitted with Adaptive Gaussian Quadrature with 15 points using the lme4 package. Multiple imputation was done using the mice package in R. The significance level used in this study was 0.05. Our study was done in accordance with the strengthening the reporting of observational studies in epidemiology (STROBE) guidelines [13].

## 3. Results

### 3.1. Patient Characteristics

A total of 11,917 severely injured patients were included in this study. These patients were admitted to 13 Level-1 trauma centers, with mean volumes varying from 120 to 410 severely injured patients per year. For the description of patients’ characteristics, hospital volume was divided in low- (<180 patients per year, N = 3532) and high- (≥180 patients per year, N = 8385) volume centers. 

Patients admitted to high-volume centers were younger (median 53, IQR 30–69 versus median 58 IQR 35–74 for lower volume centers), more frequently intubated (31.6% versus 15.6%, Table 1), in shock (6.9% versus 4.6% for low volume centers) and comatose (33.7% versus 24.3% for low volume centers). When looking at the outcome characteristics, patients admitted to high-volume centers had a longer ICU length of stay when admitted to ICU (median 4, IQR 2–9 versus median 3, IQR 2–7 for low volume centers, Table 2). Additionally, in-hospital mortality was slightly higher in high-volume centers (19.8% versus 16.4% for low volume centers, Table 2).

### 3.2. Volume-Outcome Relationship: In-Hospital Mortality

The funnel plot of volume and mortality shows small unadjusted differences (Figure 1).

There was a nonsignificant association between higher hospital volume and higher in-hospital mortality according to the unadjusted random effects model (OR 1.04 per 50 extra patients, 95% CI 0.99–1.09, *p* = 0.10, Table 3). After adjustment, there was no association between hospital volume and in-hospital mortality (OR 0.97 per 50 extra patients, 95%CI 0.90–1.04, *p* = 0.44). The correlation between the included variables varied between −0.76 and 0.77. The complete case analysis showed similar results (OR: 1.03 per 50 extra patients, 95% CI 0.94–1.12, *p* = 0.54). Sensitivity analyses showed no significant effect of hospital volume on in-hospital mortality (Table 3). 

## 4. Discussion

This study aimed to evaluate whether there was an association between hospital volume and in-hospital mortality among severely injured patients in Dutch Level-1 trauma centers. Patients admitted to the seven largest centers were younger and more severely injured compared to patients admitted to centers with smaller hospital volume. Overall, there was a small variation in hospital volume from 120 to 410 patients per year. After adjusting for differences in case mix, we did not find a relationship between hospital volume and in-hospital mortality of severely injured patients.

Compared to other countries, the variation in hospital volume in the Netherlands was small [4]. Level-1 trauma centers in the United States treat at least 240 severely injured patients per year with volumes up to 1500 severely injured patients [8,14], while the largest Dutch trauma centers treat approximately 400 severely injured patients per year [6]. This could be one of the reasons for not being able to demonstrate a significant volume-outcome relationship in our study. The Netherlands have a relatively large amount of Level-1 trauma centers compared to the number of inhabitants and surface. This creates comparable transfer times and transportation times across the country. Additionally, there are still differences in case mix among Level-1 trauma centers in the Netherlands. Patients in larger centers are more severely injured, and this is visible in the difference between unadjusted and adjusted ORs. The odds ratio goes from above one to below one after adjustment. This implies that there might be a volume-outcome relationship, but differences in case mix make it difficult to show this relationship. Additionally, Dutch trauma centers are well organized in a similar manner, which reduces the impact of volume on outcomes [15]. Dutch Level-1 trauma centers have to fulfill several criteria in order to become certified, for example, availability of several medical specialties 24/7, a minimum number of ICU beds, damage control surgery facilities and the availability of a CT-scan at the ED.

Centralization of care is suggested to improve cost-effectivity and patient outcomes [16,17]. Most evidence for the benefit of regionalization in terms of hospital volume is found in elective surgical procedures [18,19,20]. It seems likely that severely injured patients could benefit from centralization, because severely injured patients often require complex care, and having experience in treating those patients could improve patient outcomes. Specifically, in trauma care, the 24/7 availability of experienced medical staff in treating unstable patients and availability of specific specialties like neurosurgery is likely to improve outcomes of severely injured patients. On the other hand, the implementation of comprehensive trauma systems by regionalizing and standardizing complex trauma care in Level-1 facilities is likely to be more effective in improving outcomes after trauma than hospital volume itself [15]. In the Netherlands, maturation of the trauma system resulted in improved patient outcomes [3]. This was also shown in the United Kingdom, where a recent publication shows that the development of Major Trauma Networks without volume requirements increases the odds of survival for patients reaching the hospital alive [21]. Since Dutch Level-1 trauma centers need to meet various high standards, facilities and criterions, there might be little inter-hospital variation in both trauma care for outcomes and outcomes for trauma care of severely injured patients [6]. This centralization of trauma care in the Netherlands already led to a mortality reduction of 50%, and major improvements in efficiency were found [1]. 

### 4.1. Strengths and Limitations

This study has several strengths. First, we used data from the national trauma registry, which includes a detailed data collection of all admitted trauma patients in Dutch hospitals, registered by trained data managers [6]. We were also able to include patients who died due to declared intention. Some variables had a larger number of missing data, for example, respiration rate, for which we used multiple imputation as a rigorous method to handle missing data. Information about transfer and transportation times was not usable for the analysis due to the high amount of missing data. Therefore, these factors cannot be excluded as a potential confounding factor. Second, we assessed hospital volume on a continuous scale, which is statistically more efficient and leads to a more in-depth assessment of the volume-outcome relationship than categorization into low volume and high volume [22]. However, there were small differences in hospital volume between centers which could introduce a lack of power. The number of included hospitals was small because the Netherlands has only 13 Level 1 trauma centers. This could reduce the global generalizability of our results. Additionally, the possibility of Level II error does exist, although we believe that our sample size and number of included hospitals is large enough. Third, random effects logistic regression models were used instead of conservative logistic regression models. This made it possible to further adjust for random variation between Level-1 trauma centers. We saw that the effect of volume changed substantially after adjusting for case mix, so there were differences in case mix among Level-1 trauma centers. However, we cannot exclude the risk of residual confounding, since it remains difficult to adjust for differences between Level-1 trauma centers fully. In particular, not all detailed information about potential confounders like frailty (medication or cognitive performance) was available. Additionally, we were not able to look at the effect of surgeon volume on in-hospital mortality. The caseload and experience per surgeon could be important for the outcome of the patient, but previous studies showed inconsistent results [4,23,24,25]. Future studies could focus on the impact of surgeon volume, transfer status and centralization on outcome. Another limitation of our study is the lack of a good definition of the severely injured patient. The universally used injury severity measure in trauma registries and research is ISS, where ISS > 15 is defined as severely injured. However, questions about the accuracy of ISS were raised. First, an equal Abbreviated Injury Scale (AIS) in different body regions is assumed to be equal in injury severity [26,27]. Second, ISS does not account for multiple injuries in the same body region [27,28]. Therefore, it is possible that patients with equal ISS scores do not have the same injury severity, and some Level-1 trauma centers might receive more severely injured patients with equal ISS scores. Future studies should look at the proposed Berlin definition of polytrauma, which captures not only injury severity but also physiological components [29]. Lastly, we were only able to look at in-hospital mortality as an outcome, while other outcome measures like quality of life are more important for the patients surviving trauma. 

### 4.2. Implications

Our study shows that Level-1 trauma centers with a similar organization of care could potentially achieve similar results. Our findings can be extrapolated to other trauma systems and are comparable to findings from the United Kingdom [30]. Further research should include quality of life as a patient outcome, since severely injured patients suffer from long-term impairments [31]. Mortality in severely injured patients shows a decrease over the years, but high-volume centers might achieve better outcomes in quality of life. Additionally, it is important to include the workflow and treatment costs of Level-1 trauma centers. When Level-1 trauma centers become more cost-effective when they treat high volumes of severely injured patients, this could be a reason to introduce national volume cut-offs. With increasing health care costs, it is important to take cost-effectiveness into account when assessing the effect of centralization on trauma care.

## 5. Conclusions

The variation in hospital volume of the included Level-1 trauma centers was not associated with outcomes of severely injured patients. Our results suggest that well-organized trauma centers with a similar organization of care could potentially achieve comparable outcomes.

## Figures and Tables

**Figure 1 jcm-10-01700-f001:**
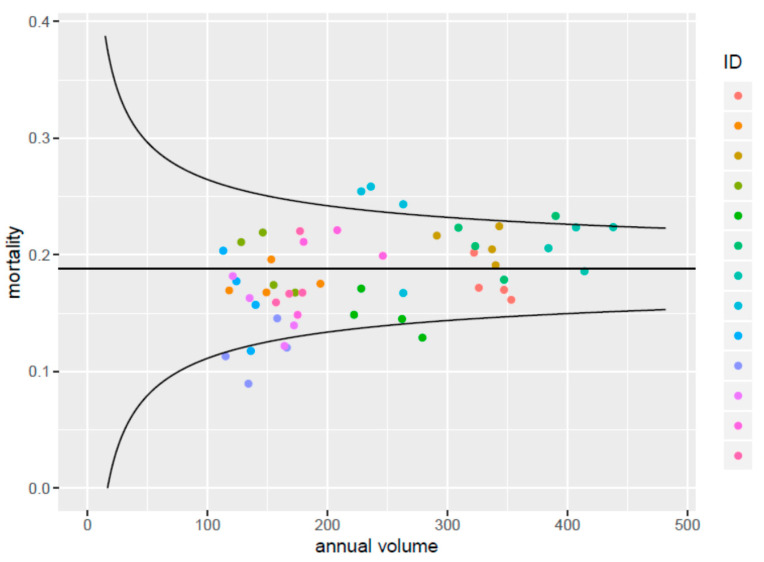
Funnel plot showing unadjusted differences in mortality rates between trauma centers, with hospital volume of severely injured patients on the x-axis. All centers are plotted for each year separately.

**Table 1 jcm-10-01700-t001:** Baseline characteristics divided by low and high hospital volume.

	Low Volume <180 Patients/Year (*n* = 3532)	High Volume ≥180 Patients/Year (*n* = 8385)	% Missings
Number of Level-1 trauma centers	6	7	0.0%
Mean hospital volume per year	147	299	0.0%
Age	58 (35–74)	53 (30–69)	0.0%
Male	2322 (65.7%)	5783 (69.0%)	0.0%
Penetrating injury	129 (3.7%)	288 (3.4%)	0.5%
ISS	22 (17–26)	22 (17–29)	0.0%
ISS > 24	1444 (40.9%)	3869 (46.1%)	0.0%
ASA comorbidities 1-A normal healthy patient 2-A patient with mild systemic disease 3-A patient with severe systemic disease 4-A patient with severe systemic disease that is a constant threat to life	1251 (48.6%) 1025 (39.8%) 293 (11.4%) 4 (0.2%)	3795 (48.9%) 2821 (36.4%) 1038 (13.4%) 103 (1.3%)	13.3%
Prehospital intubation	373 (15.6%)	2248 (31.6%)	20.3%
GCS at arrival Emergency Department (ED)	14 (8–15)	13 (3–15)	12.2%
GCS < 8	641 (24.3%)	2631 (33.7%)	12.2%
Respiration rate at arrival ED	16 (14–20)	18 (15–20)	29.6%
Systolic blood pressure at arrival ED	135 (119–157)	130 (114–150)	8.3%
Systolic Blood Pressure < 90	147 (4.6%)	535 (6.9%)	8.3%
Cardiac arrest	114 (5.2%)	542 (6.9%)	15.4%
AIS head > 3	1515 (42.9%)	3622 (43.2%)	0.0%

Continuous: median (IQR), categorical: N (%), Injury Severity Score (ISS), Abbreviated Injury Scale (AIS), Glasgow Coma Score (GCS), respiratory rate (RR), systolic blood pressure (SBP), physical status classification system (ASA).

**Table 2 jcm-10-01700-t002:** Outcome characteristics divided by low and high hospital volume (ISS > 15).

	Low Volume <180 Patients/Year (*n* = 3532)	High Volume ≥180 Patients/Year (*n* = 8385)
Length of Stay	4 (3–5)	4 (3–5)
ICU Length of Stay *	3 (2–7)	4 (2–9)
In-hospital mortality	580 (16.4%)	1660 (19.8%)

* ICU length of stay for patients with minimum of 1 day at ICU.

**Table 3 jcm-10-01700-t003:** Effect of hospital volume on in-hospital mortality.

	OR for 50 Patients Extra Per Year (95% CI)	*p*-Value
Unadjusted	1.04 (0.99–1.09)	0.10
Adjusted	0.97 (0.90–1.04)	0.44
Complete case analysis adjusted	1.03 (0.94–1.12)	0.54
**Subgroups**		
ISS > 24		
Unadjusted	1.00 (0.89–1.12)	0.97
Adjusted	0.92 (0.80–1.05)	0.22
AIS head >3 & ISS > 15		
Unadjusted	0.90 (0.75–1.08)	0.25
Adjusted	0.93 (0.79–1.09)	0.36
AIS Head <4 & ISS > 15		
Unadjusted	1.05 (0.93–1.17)	0.43
Adjusted	0.92 (0.80–1.07)	0.29

Adjusted for sex, age (spline), age × sex, co-morbidities (ASA), GCS, respiratory rate (RR), systolic blood pressure (SBP), intubation, ISS, AIS Head and penetrating injury.

## Data Availability

The data presented in this study are available on request from the corresponding author.

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
