# Peer review of "The Relationship between Hospital Volume and In-Hospital Mortality of Severely Injured Patients in Dutch Level-1 Trauma Centers"

_jcm, 2021, doi:10.3390/jcm10081700_

Round 1

Reviewer 1 Report

The authors submitted a revised version of the manuscript and point-by-point response to reviewers. Although the authors responded to some comments, they have not responded to several significant concerns adequately.

Further concerns related to multiple imputation method have been raised in the revised manuscript.

[Methods]

In the revised manuscript, the authors have included the number of incoming transfer patients in the case load counts. However, according to Figure 1, the number of patients in each hospital has not been changed at all. Please explain.

Furthermore, I believe that incoming transfer patients should be included in the analysis because the outcome of them should be evaluated as the performance of the hospital.

The authors should clearly state how multicollinearity was assessed, including the cut-off value, in the Method section. The result should be stated in the Result section.

Regarding multiple imputation, the authors should describe the number of generated data set and how they were integrated.

The package which was used for multiple imputation should also be stated.

Sensitivity analysis to validate multiple imputation method, such as complete case analysis, is generally required. 

If multiple imputation method was used, presenting patient characteristics of multiply-imputed data, in addition to naïve data, is mandatory. The data of Table 1 seemed to be naïve data. (for example, although the number of patients who received prehospital intubation in the Low volume group is shown as 373 which is not equal to 15.6%. If this is the number of multiply imputed data, 373/3532 = 10.6%)

Author Response

Dear Reviewer 1,

Thank you for your thoughtful comments. We believe it strongly improved our manuscript. Please see the point-by-point responses below:

[Methods]

In the revised manuscript, the authors have included the number of incoming transfer patients in the case load counts. However, according to Figure 1, the number of patients in each hospital has not been changed at all. Please explain.

We already included incoming transfer patients into our case load counts. But it was not clearly written down in our manuscript. We re-checked our R code and to ensure that incoming transfers were all counted as case load as you suggested. There were no changes in the case load.

Furthermore, I believe that incoming transfer patients should be included in the analysis because the outcome of them should be evaluated as the performance of the hospital.

We respectfully disagree with Reviewer 1 that incoming transfer patients should alsof be included in the analysis itself, because patients treated extensively at other hospitals could influence the results in outcome obtained by Level 1 trauma centers. For example, if a neurotrauma patient is treated at a Level 2 trauma center with neurosurgical expertise and transported to a Level 1 trauma center, the outcome might be better than when this same patient is first brought to a Level 3 trauma center without any neurosurgical facilities available. Since some regions have multiple Level 2 trauma centers and other regions do not have any Level 2 trauma centers, the outcome of transferred patients is too dependent on the hospital they were treated at first.

However, we strongly suggest that further research is done on this topic and we added this to the discussion:

"Future studies could focus on the impact of surgeon volume, transfer status and centralization on outcome. "

The authors should clearly state how multicollinearity was assessed, including the cut-off value, in the Method section. The result should be stated in the Result section.

Multicollinearity is indeed an important topic when multivariable models are used. We included this in our methods section:

"All random effects logistic regression models were evaluated for multicollinearity by assessing the correlation between the included variables. A cut-off value of -0.8 or 0.8 was used as measure of severe multicollinearity. "

We also included this in our results section:

"The correlation between the included variables varied between -0.76 and 0.77."

Regarding multiple imputation, the authors should describe the number of generated data set and how they were integrated.

The package which was used for multiple imputation should also be stated.

Sensitivity analysis to validate multiple imputation method, such as complete case analysis, is generally required. 

We agree that our description of the multiple imputation done should be more extensive. This was added to our methods section:

"For the analysis part, missing data was imputed using multiple imputation, assuming missingness at random. All variables, except for the outcome variable in-hospital mortality which had no missing values, were imputed. Continuous variables were standardized using Z-score standardization before imputation. Multiple imputation was done using the mice package in R. There were 5 imputed datasets created which were pooled using Rubin’s Rules. A complete case analysis was done to validate our results. "

The complete case analysis was added to the results and showed similar results:

"The complete case analysis showed similar results (OR: 1.03 per 50 extra patients, 95% CI 0.94 – 1.12, p=0.54). "

If multiple imputation method was used, presenting patient characteristics of multiply-imputed data, in addition to naïve data, is mandatory. The data of Table 1 seemed to be naïve data. (for example, although the number of patients who received prehospital intubation in the Low volume group is shown as 373 which is not equal to 15.6%. If this is the number of multiply imputed data, 373/3532 = 10.6%)

We included a Table 1 of dataset 1 of the multiple imputed data to our appendix. Also we included the densityplots of the continuous variables with missing values of our 5 imputed datasets to appendix B. 

Reviewer 2 Report

Thank you for the opportunity to review this work. Authors conducted this retrospective observational study to clarify the association between hospital volume and in-hospital mortality using Dutch nationwide trauma registry. This study concluded that the hospital volume was not significantly associated with in-hospital mortality after adjusting patients' characteristics. This conclusion implies the importance of not only hospital volume but also sophisticated trauma care system such as Dutch Level-1 trauma centers, which is, I consider, the important aspect of trauma care.

I think interpretation and conclusion of the results are generally appropriate. But, I'd like to point out some minor comments.

Comments to the authors:

  1. As authors mentioned in the manuscript, the variation of hospital volume between Level-1 trauma centers was small. Furthermore, this study included relatively small number of institutions (13 centers), which could reduce the generalizability of this results. Please discuss this point in discussion a little bit more.

  1. Dataset included a large number of missing data (e.g. respiration rate at arrival ED was missing in 29.6% of patients), which could influence the results of your study. Please mention this in discussion.

  1. Authors stated that the Dutch Level-1 trauma centers would provide similar level of trauma care, based on the strict certification criteria of trauma centers. Are there any data in your dataset or previous studies supporting this statement? (e.g. Are there any data about intervention performed after hospital admission?)

Author Response

Dear Reviewer 2,

Thank you for your thoughtful comments. We believe it strongly improved our manuscript. Please see the point-by-point responses below:

Comments to the authors:

  1. As authors mentioned in the manuscript, the variation of hospital volume between Level-1 trauma centers was small. Furthermore, this study included relatively small number of institutions (13 centers), which could reduce the generalizability of this results. Please discuss this point in discussion a little bit more.

This is indeed an important remark. We added this to the discussion section:

"The number of included hospitals was small because the Netherlands has only 13 Level 1 trauma centers. This could reduce the global generalizability of our results.  Also, the possibility of Level II error does exist although we believe that our sample size and number of included hospitals is large enough. "

  1. Dataset included a large number of missing data (e.g. respiration rate at arrival ED was missing in 29.6% of patients), which could influence the results of your study. Please mention this in discussion.

We added this to the discussion:

"Some variables had a larger number of missing data, for example respiration rate, for which we used multiple imputation as a rigorous method to handle missing data. "

  1. Authors stated that the Dutch Level-1 trauma centers would provide similar level of trauma care, based on the strict certification criteria of trauma centers. Are there any data in your dataset or previous studies supporting this statement? (e.g. Are there any data about intervention performed after hospital admission?)

This is an important point. One study found that centralization of trauma care in the Netherlands already led to a mortality reduction of 50% and major improvements in efficiency were found [1]. We added this to our discussion. 

1. Hietbrink, F.; Houwert, R.M.; van Wessem, K.J.P.; Simmermacher, R.K.J.; Govaert, G.A.M.; de Jong, M.B.; de Bruin, I.G.J.; de Graaf, J.; Leenen, L.P.H. The evolution of trauma care in the Netherlands over 20 years. Eur J Trauma Emerg Surg 2020, 46, 329-335, doi:10.1007/s00068-019-01273-4.

Reviewer 3 Report

Centralization of trauma has been shown to reduce the mortality of severely injured patients. In this article, the relationship between hospital volume and mortality of severely injured patients in 13 Dutch Level-1 trauma centers was evaluated. Observed mortality rates varied between 12-24% per center. After adjustment, there were no differences between low and high volume centers. (adjusted odds ratio 0.97 per 50 extra patients per year, 95% Confidence Interval 0.90-1.04). Authors conclude that well organized trauma centers with similar organization of care could potentially achieve comparable outcomes.

The article is clearly written, easy to follow. The authors have confirmed with a large sample which is still controversial issue: “Does volume of patients matter even between Level-1 Trauma Centers?” I agree that compared to US, the variation in hospital volume for some countries such as in Europe or Asia is small. This is very important topic for policy decision in regional trauma care.

Major

How does transport time and distance differ between high volume centers and low volume centers? Are the larger trauma centers associated with longer transport time/distance due to the centralization? Or, is large trauma center located in larger population density and associated with short transportation time/distance? This is important for knowing the characteristics “trauma bypass”, also, this may be considered as a confounding factor. Please, consider.

Minor

Description for statistical package is split into two. Please consider including P3, L97” Multiple imputation was done using the mice package in R.” into P3, L113-119.

In this reviewers PDF, there are 2 graphs (larger and smaller) in Figure1. Which one is verified for publication? 

Author Response

Dear reviewer,

We want to you for your thoughtful comments and suggestions. We believe that it significantly improved our manuscript. Please find our point-by-point reply and the changes made to the manuscript attached to this letter.

Kind regards,

Charlie Sewalt

-----

Major

How does transport time and distance differ between high volume centers and low volume centers? Are the larger trauma centers associated with longer transport time/distance due to the centralization? Or, is large trauma center located in larger population density and associated with short transportation time/distance? This is important for knowing the characteristics “trauma bypass”, also, this may be considered as a confounding factor. Please, consider.

You are right that longer time/distance is associated with worse outcomes in neurotrauma patients and penetrating injury patients. For haemodynamically stable trauma patients, less evidence is found that a time delay worsens outcomes [Harmsen, A. M., Giannakopoulos, G. F., Moerbeek, P. R., Jansma, E. P., Bonjer, H. J., & Bloemers, F. W. (2015). The influence of prehospital time on trauma patients outcome: a systematic review. Injury46(4), 602–609.].

We did look at the transfer times and transportation times when they were available for the patients. Since the Netherlands is a relatively small country with a large number of Level-1 trauma centers (n=13), there were no significant differences in tranportation and transfer times between the trauma centers. However, the amount of missing information was quite high (up to 80% for some trauma centers) so we decided not to analyse this further although it is very interesting.  We added this interesting topic to our discussion section:

"The Netherlands have a relatively large amount of Level-1 trauma centers compared to the number of inhabitants and surface. This creates comparable transfer times and transportation times across the country. "

"Information about transfer and transportation times was not usable for the analysis due to the high amount of missing data. Therefore these factors cannot be excluded as potential confounding factor."

Minor

Description for statistical package is split into two. Please consider including P3, L97” Multiple imputation was done using the mice package in R.” into P3, L113-119.

We summarized all information about R and the R packages used into lines 112-116 on page 3. 

"Statistical analyses were performed in R statistical software 3.5.1 (R Foundation for Statistical Computation, Vienna). Random effect models were fitted with Adaptive Gaussian Quadrature with 15 qpoints using the lme4 package. Multiple imputation was done using the mice package in R. The significance level used in this study was 0.05. Our study was done in accordance with the STROBE guidelines [13]."

In this reviewers PDF, there are 2 graphs (larger and smaller) in Figure1. Which one is verified for publication? 

We deleted the older and smaller Figure 1. Thanks for pointing this out.

Round 2

Reviewer 1 Report

I have no additional comments and congratulate the authors on their work.